# OpenReview forum: "Camouflage Patching: Effective Jailbreak Attacks on Single- and Multimodal LLMs"
_ICLR.cc/2026/Conference — ICLR 2026 Conference Withdrawn Submission_

### Official Review · Reviewer_yimJ · 2025-10-29

**Soundness:** 2
**Presentation:** 2
**Contribution:** 2
**Rating:** 2
**Confidence:** 3

**Summary:**

This paper proposes an LLM jailbreak framework designed to exploit large language models’ strong instruction-following capability and their tendency to continue generating text without reassessing the overall intent. The framework conducts jailbreak attacks by rewriting harmful queries into benign-looking requests.

**Strengths:**

- The paper focuses on LLM Jailbreaking, a timely topic.
- Clear motivations and method description.
- Evaluations are somewhat comprehensive, including possible defense methodologies.

**Weaknesses:**

The LaTeX template differs from the official version, and font consistency should be improved.

The proposed method, though effective, lacks novelty.

The evaluation is limited. Model-based ASR verification may be biased due to its black-box nature; adding rule-based checks (e.g., keyword matching) could strengthen the validation.

The paper would benefit from deeper analysis of the framework, including LLM behavior, mechanism insights, and failure cases.

**Questions:**

The whole framework is based on two important theories/assumptions:

LLMs have

- (i) strong instruction-following capability, and

- (ii) a tendency to continue following benign reconstruction steps without re-evaluating global intent.


Are there any references in validating these two properties? If not, I believe it should be proved beforehand. There are also no specific examples or case study on this.

---

### Official Review · Reviewer_Rb9r · 2025-10-31

**Soundness:** 3
**Presentation:** 3
**Contribution:** 3
**Rating:** 4
**Confidence:** 4

**Summary:**

The paper proposes CamPatch, a jailbreaking framework that leverages the instruction-following capability and compliance with benign instructions of large language models (LLMs), to construct a two-stage jailbreaking approach. CamPatch rewrites malicious instructions by replacing critical tokens with benign ones and records the mapping between the original and replaced tokens. It then structures the input as a query containing both the rewritten benign instruction and the token mapping. The query is formulated in a CoT style, prompting the model to first reconstruct the original malicious instruction using the provided mapping and then execute it, thereby bypassing safety defenses and achieving jailbreaking. The method is evaluated on multiple open-source and commercial models and compared against several baselines, demonstrating its effectiveness.

**Strengths:**

1. The paper introduces the CamPatch jailbreaking framework, which innovatively employs a two-stage approach: transforming malicious instructions into benign ones while preserving a record of the transformation. By using CoT prompting to guide the model to first reconstruct the original instruction and then execute it, the method achieves successful jailbreaking in a single query. This one-shot attack is both feasible and novel.
2. The proposed method extends to multimodal LLMs, where malicious tokens can be replaced with images, audio, or other modalities, making the malicious intent more concealed.
3. As a one-shot attack, CamPatch does not require multi-round iterations or repeated optimization. Experimental results across all tested models and baselines consistently demonstrate the method’s effectiveness.

**Weaknesses:**

1. The experimental evaluation lacks testing on the latest and very large open-source models (e.g., 70B), limiting the demonstration of the method’s generalizability. Including results on more recent large models would strengthen the paper’s claims.
2. As shown in Table 2, the method appears to have limited effectiveness against more capable models (e.g., Claude 3.5 Sonnet).
3. The experiments do not include reasoning-focused LLMs, making it difficult to assess the method’s broad applicability and robustness across different model architectures.
4. The proposed potential defense mechanisms lack experimental data support. In particular, Section 7.3 mentions “Adaptive Defense,” but there is no experimental evidence showing whether CamPatch can effectively bypass such defenses.

**Questions:**

1. The experiments do not evaluate reasoning LLMs. I am curious whether such models, during their internal reasoning process prior to generating a final response, would detect that the reconstructed instruction is malicious and thus refuse to comply. The authors could strengthen their method by including test results and analysis on reasoning LLMs to demonstrate the method’s broader applicability.
2. Although the CamPatch-crafted query appears benign on the surface, it explicitly includes the token mapping table that reveals the original malicious tokens. Can CamPatch truly bypass input-level keyword-based filtering defenses (e.g., simple grep-style keyword scanners)? Section 7.1 does not address such keyword-based filtering mechanisms, raising questions about robustness against this common defense.
3. See weaknesses above.

---

### Official Review · Reviewer_pETR · 2025-11-01

**Soundness:** 4
**Presentation:** 4
**Contribution:** 2
**Rating:** 4
**Confidence:** 4

**Summary:**

The paper presents a well-motivated and thoroughly evaluated attack method (CamPatch) that meaningfully advances our understanding of LLM jailbreaks. Its core idea—rewriting malicious prompts into innocuous queries with explicit reconstruction cues—is interesting, and the experimental gains over prior techniques are convincing.

**Strengths:**

1. This paper provide clear motivation and rationale for the proposed CamPatch attack method.
2. Massive experiments are conducted to validate the effectiveness of CamPatch across various models and settings.
3. The presentation of this paper is clear and well-structured.

**Weaknesses:**

1. **Incremental Conceptual Contribution:** While the authors convincingly illustrate the rationale behind CamPatch and its empirical effectiveness, the core idea of rewriting malicious prompts into innocuous queries with reconstruction cues is somewhat incremental. Similar practices have been explored in prior jailbreak work [1], which also employs disguise and reconstruction strategies. Although there exist differences in the specific techniques and implementation details, I am afraid that the essential concept of CamPatch is not significantly distinct from existing methods. The authors are encouraged to further clarify their contributions, both conceptually and technically, that distinguish CamPatch from prior work.
2. **Naturalness of Queries:** The attack’s query format (e.g. “First replace placeholders using the mapping; then answer the question”) may itself be unnatural or suspicious as user input. It’s unclear if real users would pose requests in this form, and if filters or human reviewers might flag such stepwise instructions. Further, there may existing defenses that specifically target such reconstruction cues. The authors are encouraged to discuss the realism of the attack queries and potential mitigations.
3. **Selection of Baselines:** The baselines considered in the experiments are somewhat limited. While the authors compare against several prior jailbreak methods, it would strengthen the evaluation to include more recent and advanced techniques, especially those leverageing chain-of-thought or multi-turn interactions, e.g., Crescendo. This would help validate the relative effectiveness of CamPatch against a broader spectrum of jailbreak strategies.


Ref
1. Liu, Tong, et al. "Making them ask and answer: Jailbreaking large language models in few queries via disguise and reconstruction." 33rd USENIX Security Symposium (USENIX Security 24). 2024.

**Questions:**

1. What is the essential contribution of CamPatch compared to prior jailbreak methods using disguise and reconstruction strategies?
2. How does CamPatch compare against more recent jailbreak techniques, such as chain-of-thought or multi-turn attacks?
3. How realistic are the attack queries in practice, and could they be detected or mitigated by heuristic filters?

---

### Official Review · Reviewer_KAtf · 2025-11-01

**Soundness:** 3
**Presentation:** 3
**Contribution:** 3
**Rating:** 6
**Confidence:** 3

**Summary:**

The paper proposes Camouflage Patching (CamPatch), a jailbreak technique that rewrites a malicious prompt into a benign-looking one using placeholder substitutions, then asks the model to reconstruct and execute the malicious query in one shot. The attack exploits the tendency of aligned models to follow structured procedural instructions and not re-run safety filters once inside a chain-of-thought–style transformation. The method is tested on both text-only LLMs and multimodal LLMs, demonstrating higher attack success rates than several prior prompt-based and obfuscation-based jailbreaks.

**Strengths:**

1. Works in a single query, which matters for real-world black-box scenarios.
2. Empirical results cover both open-source and commercial models.
3. Explicit evaluation on multimodal models, with concrete mechanisms to map harmful tokens into image/emoji placeholders.
4. Ablations indicate the importance of the two-stage rewrite + reconstruction and CoT/ICL components.
5. Shows that many aligned systems still break when malicious intent is revealed late in a reasoning trace.

**Weaknesses:**

1. While the idea is framed as camouflage + rule reconstruction, it is essentially a structured prompt-rewriting recipe rather than a fundamentally new jailbreak paradigm. Most components—benign paraphrasing, placeholder substitution, and CoT-guided reveal—have appeared in pieces in prior work. The contribution feels incremental in technique despite strong results.
2. The approach uses explicit templates and few-shot mapping exemplars, rather than learning-based or more generalizable mechanisms. It’s unclear how robust this is to adversarial defense prompts, alternative model instruction formats, or system-prompt hardening.
3. ASR uses a GPT-based judge with "specificity" and "convincingness" scores. While common in jailbreak benchmarking, the paper would benefit from reporting exact harmful content categories and qualitative failure cases. The current metric makes it hard to tell if outputs are genuinely dangerous or merely plausible-sounding.
4. Although the paper substitutes tokens with emojis/images for MLLMs, the attack still fundamentally operates via language-based reconstruction, not genuine cross-modal logic exploitation. It’s unclear whether the visual tokens meaningfully affect safety behavior beyond serving as surface obfuscation.
5. The paper briefly tries self-reminders and CoT-based defenses, but does not examine training-time or system-level defenses. Suggesting "train on these attacks" is somewhat obvious; more principled discussion of mitigation would strengthen the work.

**Questions:**

1. How sensitive is the attack to the exact prompt template and ordering rules?
2. Does the attack survive context-length truncation or noisy system prompts?
3. For MLLMs, how much does the image token matter vs. just a neutral text token?
4. Have you tested models with continuous alignment defenses (e.g., internal self-critique layers)?

---

### Note · Authors · 2025-12-03

**Comment:**

I have read and agree with the venue's withdrawal policy on behalf of myself and my co-authors.

**Withdrawal Confirmation:**

I have read and agree with the venue's withdrawal policy on behalf of myself and my co-authors.